# Understanding the Role of the Glial Scar through the Depletion of Glial Cells after Spinal Cord Injury

**DOI:** 10.3390/cells12141842

**Published:** 2023-07-13

**Authors:** Lucila Perez-Gianmarco, Maria Kukley

**Affiliations:** 1Achucarro Basque Center for Neuroscience, 48940 Leioa, PC, Spain; maria.kukley@achucarro.org; 2Department of Neurosciences, University of the Basque Country, 48940 Leioa, PC, Spain; 3IKERBASQUE Basque Foundation for Science, 48009 Bilbao, PC, Spain

**Keywords:** glial scar, spinal cord injury, cell ablation

## Abstract

Spinal cord injury (SCI) is a condition that affects between 8.8 and 246 people in a million and, unlike many other neurological disorders, it affects mostly young people, causing deficits in sensory, motor, and autonomic functions. Promoting the regrowth of axons is one of the most important goals for the neurological recovery of patients after SCI, but it is also one of the most challenging goals. A key event after SCI is the formation of a glial scar around the lesion core, mainly comprised of astrocytes, NG2^+^-glia, and microglia. Traditionally, the glial scar has been regarded as detrimental to recovery because it may act as a physical barrier to axon regrowth and release various inhibitory factors. However, more and more evidence now suggests that the glial scar is beneficial for the surrounding spared tissue after SCI. Here, we review experimental studies that used genetic and pharmacological approaches to ablate specific populations of glial cells in rodent models of SCI in order to understand their functional role. The studies showed that ablation of either astrocytes, NG2^+^-glia, or microglia might result in disorganization of the glial scar, increased inflammation, extended tissue degeneration, and impaired recovery after SCI. Hence, glial cells and glial scars appear as important beneficial players after SCI.

## 1. Glial Scar Formation and Composition

Spinal cord injury (SCI) is a condition that affects between 8.8 and 246 people in a million, according to the World Health Organization. Unlike many other neurological disorders, SCI affects mostly young people, causing deficits in sensory, motor, and autonomic functions. Axonal atrophy following injury to the central nervous system (CNS) and the inability of central axons to regenerate has been described already by Ramón y Cajal in 1928 [1], but to this day, promoting regrowth of injured axons remains one of the main challenges to regain function. Various factors may affect axonal regeneration after SCI and one of them is the glial scar, which has been proposed by several studies as the main constraint for repair [2,3,4].

The glial scar is a compact mesh structure that surrounds the CNS lesions rapidly after the initial damage and is fully formed between two and three weeks post-injury (wpi) in rodents. This process is triggered by the breakdown of the blood-spinal cord barrier (BSCB) and the infiltration of non-neural cells into the CNS parenchyma [5]. Early on, this infiltrated parenchyma in the epicenter lesion presents edema, myelin debris, degenerating neurons and glia. As days go by, the edema is reduced, and a fibrotic scar begins to form, occupying the entire lesion epicenter by eight days post-injury (dpi) in rodents [6]. The fibrotic scar is composed of invading hematogenous cells: lymphocytes, macrophages, and leukocytes that infiltrate the spinal cord parenchyma; additionally, days after injury, fibroblasts from the damaged meninges invade the lesion core, where they proliferate and secrete molecules that change the extracellular matrix (ECM) [7]. Glial cells are rarely found within this fibrotic tissue, accumulating around it instead and separating the lesion epicenter from the spared nervous tissue by forming the glial scar (Figure 1).

This glial scar is mainly composed of astrocytes, NG2^+^-glia, and microglia, with newly proliferated astrocytes playing the leading role [7] (Figure 1). In human tissue, neurons, and oligodendrocytes are the vast majority of apoptotic cells, found between 3 h and 2 months post-injury [8]. At 1 dpi, microglia have already responded and changed their morphology, and they can remain at the site of injury for weeks to months [9]. It takes a few days for astrocytes near the lesion to undergo hypertrophy, reaching the maximum at 2–3 wpi and generates the glial scar [9]. This series of events is similar to the ones occurring in the animal models, although timing is the main difference. In rodents, astrocytes can be identified in the vicinity of the lesion epicenter by 2 dpi, where they progressively accumulate to form the dense structure that isolates the lesion epicenter from the spared tissue, with their processes extending towards the lesion [6]. During the first two weeks after SCI in rodents, astrocytes, NG2^+^-glia, and microglia undergo extensive proliferation, and their numbers remain elevated during the chronic phase [10]. Other than their accumulation and dense organization, these glial cells undergo diverse functional and morphological changes, affecting the composition of the extracellular matrix (ECM); this phenomenon is common to rodents and humans (Figure 1).

The ECM is an essential, non-cellular scaffold that varies for each tissue type but is always made up of collagens, fibronectin, laminin, hyaluronan, and elastin, added to different proteins, proteoglycans, and glycosaminoglycans [11]. The composition of the ECM can influence cell adhesion, migration, proliferation, and differentiation, and therefore, cellular and tissue function [12]. In the healthy adult CNS specifically, the ECM is looser than in other tissues because it is constituted primarily by hyaluronan, sulfated proteoglycans, and tenascin-R [13,14]. The molecules comprising the ECM play a role in axonal guidance and synaptogenesis that can vary between regions, ages, and animals [15,16]. For example, tenascin-R inhibits axon growth in mouse cerebellar neurites [17], but it promotes growth in embryonic mouse hippocampal neurons [18]. In the post-SCI, glial and blood-derived cells in the lesion overexpress matrix metalloproteases (MMPs) [19], resulting in the degradation of hyaluronan and other ECM components into low molecular weight fragments that can amplify inflammation [20]. Additionally, glial cells modify the ECM composition further by overexpressing sulfated proteoglycans, among them chondroitin sulfate proteoglycans (CSPGs) like NG2, brevican, neurocan and versican, both in rodents [21,22,23,24] and humans [25]. The possible consequences of sulfate proteoglycans overexpression include restriction of axon plasticity, migration of inflammatory cells, sequestering or presenting growth factors, cytokines, and chemokines (Reviewed in [19]). Dysregulation of these sulfated proteoglycans can have long-lasting effects on the structure and dynamics of the local tissue [19] and represents one of the major factors inhibiting axonal regeneration [26]. This is one of the main reasons why the glial scar has traditionally been viewed as detrimental to recovery after SCI. However, a growing number of experiments demonstrate that the glial scar plays a neuroprotective role as well [27,28]. In this context, the depletion of the glial scar, or the cells involved in its formation, represents interesting approaches to determine whether the glial scar promotes or inhibits axonal regeneration and functional recovery after SCI. Here, we review the studies that specifically ablated each glial cell population involved in the glial scar formation and describe how those manipulations affected recovery after SCI in animal models (Table 1).

## 2. Conditional Cell Ablation Strategies

### 2.1. Genetic Approaches for Ablation of Specific Glial Cell Populations

Several genetic strategies of conditional cell ablation have been described in the literature. Their major principle is the expression of a foreign protein, typically of bacterial or viral origin, whose transcription is regulated by a promoter specific to the targeted cell population. Such protein, be it an enzyme or a receptor, will grant that cell population the unique capacity to break down an administered prodrug into its active form, initiating the cell’s apoptotic death through one of several possible mechanisms.

One of these cell ablation strategies is the Tk/GCV system, where the Thymidine kinase (Tk) gene -of the Herpes Simplex Virus- is expressed under the control of a cell-specific promoter. Then, when the Ganciclovir (GCV) prodrug is applied systemically, the Tk (expressed exclusively in the cells of interest) phosphorylates it to its active form, inhibiting DNA synthesis and leading to apoptosis of the proliferative population of interest, and thus to its elimination (Figure 2). This system can be used in transgenic animals that express the Tk under the cell-specific promoter of interest, or, alternatively, it can be delivered through viral transfection. However, given its mechanism of action, the Tk/GCV system can only ablate proliferating cells.

Other similar systems that can target non-proliferating cells include (1) the NTR-CB1954, in which the *E. coli* enzyme nitroreductase (NTR) activates the CB1954 prodrug, inducing DNA interstrand cross-linking and cell death in the cells expressing NTR [35,36,37]; and (2) DTA/DTR, where a Diphtheria toxin receptor (DTR) is expressed under a cell-specific promoter and is harmless until the moment of ligand (Diphtheria toxin A, DTA) administration. When DTA binds to the DTR, the inhibition of protein synthesis is triggered in the targeted cells, leading to their apoptosis [38].

A common disadvantage to these cell ablation strategies is the necessity to obtain and maintain (or even generate new) transgenic mice that express the involved enzyme or receptor under a cell-specific promoter. This may be expensive and time-consuming. Additionally, there might be small populations within a determined cell type that escape depletion by expressing the wild-type allele and failing to express the Cre and reporter protein. Such a phenomenon has been recently described for microglia in the Cx3cr1^CreER-Eyfp/wt^ mouse, where less than 1% of the population is Cx3cr1^+^ Cre^−^ Eyfp^−^, thus failing to be targeted for ablation [39]. Although the same issue has not been reported for the other transgenic models addressed in this review, it should be considered a possibility when interpreting results. Alternatively, other available approaches for ablation of glial cells can be considered, e.g., pharmacological.

### 2.2. Pharmacological Approaches for Ablation of Specific Glial Cell Populations

Pharmacological approaches employ drugs or toxins that affect a specific cell type and can be applied to non-transgenic laboratory animals. However, the risk of using such toxins or drugs is the possible off-target effects. For example, in the case of microglia, it has been shown that these cells are the main source of Colony Stimulating Factor 1 Receptor (CSF-1R) expression in the CNS, and CSF-1R ablation or loss of function leads to microglia depletion [40]. Thus, it is reasonable to expect that inhibition of CSF-1R can lead to conditional microglial depletion. In fact, several specific kinase inhibitors have been developed and tested in vivo, some efficiently crossing the BBB and ablating microglia [41]. Among them, PLX3397 and PLX5622, which can be administered orally in the chow, appear to be the most potent and are widely used in microglia research [31,32,33,41,42,43]. Moreover, the Food and Drug Administration (FDA) has approved PLX3397 as a therapeutic agent for patients with Tenosynovial giant cell tumors, where blocking the CSF-1/CSF-1R pathway reduces the number of infiltrating monocytes and macrophages and minimizes tumor growth [44].

For specific ablation of astrocytes, the typical toxin is L-alpha-aminoadipate (L-AAA). L-AAA is a naturally occurring glutamate homolog that constitutes an intermediate in the lysine metabolism [45,46]. The L-AAA inhibits glutamate synthesis [47] and uptake [48] in astrocytes, rapidly ablating them without affecting the surrounding cell types, even if administered for extended periods; however, it is incapable of crossing the BBB in the adult CNS and must, therefore, be injected into the brain or spinal cord [48].

Even though there are pharmacological drugs that induce oligodendrocytes death, such as the *Clostridium perfringens* Epsilon toxin [49] and Cuprizone (effect and mechanism of action reviewed by Zirngibl et al. [50]), no drugs for specific ablation of NG2^+^ oligodendrocyte precursor cells have been reported so far.

Regardless of the chosen method to ablate glial populations, some considerations are always pertinent. Deletion of a group of cells might have an effect on its own -regardless of the targeted population- by causing inflammation and infiltration, possibly mediating some of the observed changes. In order to clarify the size of this effect, controls can be included in the experiments, such as sham-operated animals that undergo ablation and are, therefore, expected to represent the basal inflammation of the model. Additionally, lipopolysaccharide (LPS) injections induce acute inflammation [51] and have been used to differentiate between the effects of inflammation and the proper effects of eliminating the glial cell type [29].

Moreover, Zhou et al. found a microglial population that persisted after conditional ablation with a genetic approach but also described that PLX3397 treatment for 21 days ablated approximately 70% of the total microglia [39]. In contrast, Fu et al. reported that PLX3397 treatment for seven days resulted in a 95% reduction in the Iba1^+^ population [33]. These variations in treatment efficiency might result from different experimental designs, age, and/or strain of the animals; hence, it is advisable to confirm the effect of the chosen treatment in the experimenter’s hands in order to conclude accordingly about the obtained data.

## 3. Glial Scar after SCI: What Have We Learned from Astrocytic Ablation Experiments?

Astrocytes are glial cells that have multiple functions in the adult CNS, including regulation of the BBB and CSF-brain barrier formation and function, regulation of blood supply, uptake of glucose, glutamate, and lactate supply to neurons, regulation of synaptic transmission and plasticity, pruning, and maintenance of synapses, removal of neurotransmitters from the extracellular space, extracellular ions homeostasis and pH control. Embryonic astrocytes also support axonal growth, but this function gets lost with age until the point when mature astrocytes inhibit axon growth [52,53].

In the intact adult spinal cord, astrocytes barely proliferate, but damage to the CNS triggers an astrocytic response known as astrogliosis. During this process, inflammatory cytokines promote the proliferation of adult astrocytes and ependymal cells, giving origin to reactive astrocytes [54,55]. In rodent models of SCI, first signs of astrogliosis appear as early as 2–7 dpi [10,29], but reactive astrocytes persist chronically after SCI. These newly proliferated reactive astrocytes -which acquire a different morphological structure with extended thick processes- are the ones that comprise the glial scar, where their cell density nearly doubles compared to astrocytes in uninjured tissue [56].

One of the main reasons why astrocytes (and the whole glial scar) are viewed as detrimental to axon regrowth is their overexpression of CSPGs after SCI [24,57], which is associated with inhibition of axonal regeneration [58,59]. However, other cell types in the scar also express CSPGs [22,60,61], and astrocyte ablation does not eradicate CSPGs in the lesion [21]. Most importantly, some of those CSPGs are axon-growth permissive; therefore, CSPG presence in the glial scar does not prevent axon regrowth in a stimulated environment [21]. The role of CSPGs in the glial scar after SCI is just one of the interesting findings resulting from astrocytic ablation experiments.

Astrocytic ablation was first described in forebrain stab injuries by means of the Tk/GCV system in adult transgenic mice [62]. Since then, several groups have made use of this system to ablate reactive proliferating astrocytes in mouse models of SCI, either by using transgenic *GFAP-Tk* mouse lines [21,29] or by lentivirus-mediated gene expression [30]. Both strategies proved effective at ablating this cell population, although the lentiviral approach did affect a small number of Olig2^+^ cells [30]. Ablation of scar-forming astrocytes resulted in extended tissue damage, failure to repair the BSCB and the consequent augmented infiltration of the spinal cord parenchyma by blood-borne cells, more extensive demyelination and oligodendrocyte death, decreased neuronal survival near the injury, and diminished locomotor function recovery [29,30]. These results were observed at different time points ranging from two to six weeks post-injury (wpi).

Anderson et al. used the same approach to ablate reactive proliferating astrocytes but focused their study on axonal regrowth in three fiber tracts: the descending corticospinal tract (CST), the ascending sensory tract (AST) and the descending serotonergic tract (5HT) [21]. They report that not only did astrocyte ablation not promote spontaneous axonal regrowth, but it increased the axonal dieback in the AST and CST. The 5HT axons in mice were less affected by the SCI and did not present augmented dieback after astrocytic ablation; still, they failed to spontaneously regrow in the lesion center at 8 wpi.

Seeing that astrocyte ablation in the acute phase of SCI in mice can be detrimental, Anderson et al. ablated the astrocytes during the chronic phase of SCI (i.e., 5 wpi), making use of a DTA/DTR system [21]. At 10 wpi, the authors observed failure of the axons to spontaneously regrow within all three fiber tracts studied. Interestingly, if neurotrophin-3 (NT3) and brain-derived neurotrophic factor (BDNF) (factors promoting axon growth) were administered locally after SCI, few AST axons were found to regrow into the lesion core [21]. However, administration of these same neurotrophic factors in combination with preconditioning lesions (a process in which the sciatic nerve is transected and ligated one week before SCI) [63,64,65] successfully promoted AST axon regrowth through the glial scar, despite CSPG presence [21]. Tk/GCV ablation of reactive astrocytes in this paradigm completely prevented axon regrowth, indicating that astrocytes are not detrimental to axon regeneration after SCI, but support it [21].

Overall, astrocytic ablation results in extended damage. Increased excitotoxicity seems an expected consequence, given their role in clearing molecules from the extracellular space. Additionally, their elimination enhances the extravasation of blood-borne cells, which are also known to inhibit regrowth. Consequently, axon regrowth was not enhanced, but dieback was extended. Under proper stimulation, axon regrowth occurred only in the presence of astrocytes. Taken together, astrocyte ablation experiments prove that reactive astrocytes mainly play a beneficial role after SCI and may be important for recovery at the cellular and functional levels, even at chronic stages of SCI. Hence, therapies to promote axonal regeneration should not aim at eliminating the astrocytic scar altogether but, possibly, at finding a way to reduce scarring by controlling the inflammation and infiltration of blood-borne cells.

## 4. Microglia Ablation after SCI: How Does It Affect Glial Scar Formation?

Even though for years -and to this day- microglial cells have been categorized as “resting” or “active”, they are, in fact, the most dynamic cells in the healthy CNS [66]. Even under normal conditions, microglia throughout the whole nervous tissue are always actively sensing their environment [67] and rapidly responding to its changes [68]. Among the key functions of microglia in the healthy adult nervous system are surveillance, phagocytosis of apoptotic neurons, pruning of synapses, neurotransmission regulation, and circuit reorganization (reviewed by Tremblay et al. [69]).

In response to focal CNS lesions in rodents, microglia can rapidly reorient their processes toward the lesion, and already after 30 min, they form a border that seals the lesion and prevents the spreading of the damage [67,70]. After SCI, microglial response is complex: cells retract their cytoplasmic processes and become more circular in shape [31], they proliferate [71], produce pro-inflammatory cytokines [72] and reactive nitrogen species [73], and enter a highly phagocytic state [31]. At 14 dpi in rodents, microglia accumulate in the lesion and can be found between the fibrotic and glial scars contacting astrocytes, where they persist at least until 35 dpi [31] (Figure 1).

In the last five years, a different microglial state has been described in models of pathologies such as Alzheimer’s disease [74,75], multiple sclerosis [76], demyelination [77] and, most recently, spinal cord injury [78]. Baseline microglia present transcriptional changes 0–2 h after injury, passing through an intermediate state that lasts up to one week before acquiring the “disease-associated microglia” transcriptional identity. This phenotype is maintained chronically after spinal cord injury and presents a similar transcriptional profile to embryonic and postnatal microglia in the developing CNS [78].

Microglial ablation in rodent SCI models has been carried out pharmacologically with either PLX5622 [31] or PLX3397 [32,33]. In all those experiments, microglia ablation resulted in a transient enlargement of the fibrotic scar area at 7 dpi that was no longer visible from 14 dpi onward. However, Bellver-Landete et al. found secondary satellite lesions filled with blood-derived myeloid cells and pericytes at 14 and 35 dpi [31]. Other early effects of microglial ablation were a reduction in astrocytic proliferation at 7 dpi [31,32] and decreased locomotor recovery from 7 dpi onward [31,32,33]. All these studies agree that after 14 dpi, the astrocytic scar of microglia-ablated mice is less compact, and astrocytes are not aligned but randomly positioned around the lesion center instead. This is accompanied by an exacerbated infiltration of blood-derived cells into the spinal cord parenchyma and a reduced number of neurons near the lesion. In addition to these effects, Bellver-Landete et al. observed a decrease in the Olig2^+^/CC1^+^ oligodendrocyte population around the lesion [31].

Zhou et al. quantified proteins after microglial ablation and found increased levels of three pro-inflammatory cytokines, i.e., TNF-α, IL-6, and IL-1β, which could be responsible for the augmented neuronal cell death after SCI [32]. Fu et al. analyzed the dieback of CST axons after SCI in mice with and without microglia ablation [33]. The authors injected biotinylated dextran amine (BDA)—a tracer that allows axon visualization—into the cortex of mice at 6 wpi. Two weeks later (8 wpi), they obtained sections from the injured spinal cords, stained them to see the BDA-labelled axons, and found that mice with microglia ablation presented more extended axonal dieback further away from the lesion site.

Additionally, Zhou et al. obtained transcriptomes from the lesion and adjacent spinal cord tissue at 7 dpi after SCI in mice treated or untreated with PLX3397, as well as from the sham-operated control animals [32]. They found upregulation of several groups of genes in the SCI group vs. sham-operated controls, specifically, genes involved in inflammation, fibrosis, extracellular matrix organization, cell adhesion, and immune response. The PLX3397 treatment reverted many of these changes after SCI, presenting a genetic expression more similar to uninjured mice; this indicates that microglial ablation prevents the inflammatory and immune responses initiated after SCI.

Given the negative effects of microglial depletion on SCI recovery, Bellver-Landete et al. used an additional interesting strategy to study the role of microglia after SCI [31]. The authors aimed to enhance microglial proliferation at the injury site by using a local injection of a bioresorbable hydrogel with incorporated macrophage-colony stimulating factor (M-CSF). The M-CSF is a cytokine that induces the proliferation and maturation of macrophages and microglia in rodents [79] and humans [80]. The proliferative effects of M-CSF are dose-dependent [79] and immediate on human cultured microglia [80]. Although the proportion of proliferative cells may vary, microglial division occurs consistently during treatment [80]. The hydrogel can sustain drug release for three to seven days, therefore, this system allows an increase in the proliferation of microglia during this time period. Stimulation of microglia proliferation resulted in reduced lesion area at 7 dpi and better locomotor recovery between 7 and 21 dpi, further supporting the idea that microglia exert a protective role after SCI.

Taken together, the extensive evidence gathered by several research groups unequivocally revealed microglia as a cell population whose implication after SCI is purely beneficial. Whether such a protective role is a unique property of disease-associated microglia or is common to all microglial states is an interesting unanswered question.

Finally, Han et al. have recently reviewed the existence of sexual dimorphism in rodent microglia, indicating that microglial response to SCI can differ between males and females [81]. Interestingly, of the articles reviewed in this section, both Fu et al. and Zhou et al. performed the experiments using females exclusively [32,33], while Bellver-Landete et al. pulled male and female mice [31]. Although they all report similar effects at the injury level, it would be interesting to figure out if there are differences in the response at a cellular level. Most cases of SCI occur in male patients, therefore, knowing if such dimorphism exists in humans and affects the response to this pathology in particular are important questions to address in the future.

## 5. NG2^+^-Glia Ablation after SCI: What We Learnt and What Remains Unknown

NG2^+^-glia constitutes the fourth most abundant glial cell type and the most widely distributed proliferating cell type in the normal adult CNS [82]. Differentiation of NG2^+^-glia cells into oligodendrocytes constitutes their canonical function, which is maintained throughout life [83]. These cells are capable of proliferating, migrating, and differentiating in response to external molecules, which they can rapidly sense with their motile processes [84,85]. One of the most remarkable features of NG2^+^-glia is their synaptic communication with neurons [86,87], which, together with their morphological complexity and persistence in the CNS throughout life, suggests that NG2^+^-glia may play other roles in the healthy CNS, independently of its capacity to generate oligodendrocytes [88].

NG2^+^-glia show a wide variety of responses to CNS injury, but the functional role of these responses is only partially understood and might vary with different factors, including the cause of injury, its location within the CNS, the age of the animals, etc. One type of NG2-glia response is injury-induced cell-fate plasticity, where they not only differentiate into oligodendrocytes to promote remyelination of axons [89] but can additionally differentiate into Schwann cells after focal demyelination [90], and into astrocytes after SCI [91]. Other types of NG2^+^-glia response to CNS injuries include their activation and participation in the glial scar formation after SCI [25] and promoting inflammation after injury to the brain white matter [92].

After SCI, NG2^+^-glial cells migrate towards the injury site [85] and undergo massive proliferation, representing 50% of all proliferating cells [93] and reaching their maximum at 7 dpi [34]. Even though their numbers are increased, NG2^+^-glia are incapable of populating the fibrotic tissue formed in the lesion epicenter but rather remain within the glial scar arising around the lesion [94,95,96]. Being a target of various factors which can influence their fate -such as bone morphogenetic proteins (BMPs, growth factors) produced by reactive astrocytes [97] or overexpressed pro-inflammatory cytokines (reviewed by Levine [98])- NG2^+^-glial cells may differentiate into astrocytes after SCI [99,100]. At the same time, another part of the NG2^+^-glia population may differentiate into Schwann cells and oligodendrocytes [101]. The newly differentiated Schwann cells and oligodendrocytes play a key role in remyelination, as many oligodendrocytes originally present in the tissue die, while the spared ones are incapable of remyelinating the axons [102,103]. It should be noted that the relevance of remyelination after SCI in mice is still a matter of debate. Several studies have reported improved locomotor recovery after transplantation of neural precursor cells (NPCs) capable of myelination [104,105,106], suggesting that remyelination may be important for neuroregeneration after SCI. However, in addition to remyelination, transplanted NPCs may induce change by differentiating into astrocytes or by providing trophic support to axons; thus, it is possible that the observed recovery is mediated by a combination of these effects rather than by remyelination itself [107,108]. Furthermore, Duncan et al. analyzed the effect of knocking down the Myelin regulatory factor (*Myrf*) and found that, even though Myrf is essential to induce remyelination, its knockdown does not affect locomotor recovery [109]. These findings indicate that remyelination is not required for locomotor recovery after SCI in mice.

Another type of response of NG2^+^-glia after injury is CSPGs overexpression. Although evidence exists that CSPGs (and NG2 among them) are inhibitory for axon regrowth after SCI [110,111,112], several studies have observed that regenerating axons in the injured spinal cord appear more frequently in areas of the spinal cord showing NG2 labeling [24,113]. This suggests that the NG2 proteoglycan may actually provide a permissive environment for axons to grow. However, even if this is the case, NG2^+^-glia can still entrap the growing axons through the establishment of NG2^+^-glia-axon synapses, preventing their regrowth after SCI [114,115,116].

Given the several, and sometimes opposing, roles that the NG2^+^ glia play after SCI, ablating them in vivo seems a good approach to better define if their overall function is detrimental or beneficial in the post-SCI environment. In order to ablate proliferating NG2^+^ cells, Hesp et al. made use of a Tk/GCV system with transgenic *NG2-Tk* mice and GCV infusion starting immediately after SCI [34]; the animals were sacrificed, and the tissue was analyzed at 7, 11 and 21 dpi. The animals sacrificed at 7 and 11 dpi received GCV infusion for 7 and 11 days, respectively, while the animals sacrificed at 21 dpi received GCV infusion for 14 days and were kept for an extra week to study the NG2^+^ cells response after the ablation treatment was stopped [34]. Notably, NG2^+^ is expressed not only by oligodendroglial progenitors but also by pericytes, and the used approach eliminates both cell populations. However, after SCI, NG2^+^ pericytes occupy the fibrotic scar formed at the epicenter of the lesion, while NG2^+^ oligodendroglial progenitors occupy the glial scar around the lesion. Therefore, the effects of abolishing each cell type can be studied and distinguished in the respective area of interest. Additionally, the authors performed a time course study to determine the ratio of proliferating NG2^+^ glia and pericytes throughout the first 14 dpi by staining spinal cord sections with Ki67, NG2, and PDGFRb antibodies [34].

NG2^+^ ablation resulted in loss of tissue integrity, extended edema, and hemorrhage in the lesion site at 7 dpi. The effects were still visible -although less pronounced- at 11 dpi when lesion expansion was also observed. At 21 dpi, the swelling and the signs of edema were significantly reduced and were similar to sham-operated control mice [34]. Regarding the glial scar formation, NG2^+^ cell ablation resulted in a more discontinuous and less compact glial scar, with reduced astrocyte numbers and increased macrophage infiltration. Moreover, at 21 dpi (7 days after stopping GCV treatment) presence of NG2^+^ cells reached control levels, and axons were penetrating the lesion epicenter in the Tk/GCV treated mice; these axons were usually observed in contact with NG2^+^ cells, some even seemingly “traveling along NG2^+^ processes”. This suggests that acute ablation of NG2^+^ oligodendroglial progenitors and/or NG2^+^ pericytes results in a more permissive environment for axonal regrowth afterward. However, at the same time, NG2^+^ cell ablation negatively affected locomotion, impairing recovery of the forelimb step length in the Tk/GCV treated mice, as opposed to the complete recovery observed in control mice with SCI at 21 dpi [34]. It would certainly be interesting to extend the period of analysis after stopping GCV treatment in order to see whether the glial and fibrotic scars recover to their full density after a longer NG2^+^ repopulation period and what that may mean in terms of axon regrowth and locomotor recovery.

Taken together, although experiments with ablation of NG2-expressing cells brought interesting results, they did not allow us to fully understand the role of NG2^+^-glia in the glial scar formed after SCI. The beneficial effects of NG2^+^ cell ablation could be linked not only to NG2^+^-glia but also to the changes in the fibrotic scar after ablating the NG2^+^ pericytes. Therefore, finding a way to ablate the NG2^+^-glia exclusively, without affecting other cell types expressing NG2 could be a major breakthrough in identifying the function of the NG2^+^-glia in the SCI environment. One way to do this may be designing a Split-Cre system [117] under the control of two NG2^+^-glia specific promoters (such as NG2 and Olig2) that may enable the analysis of NG2^+^-glia functions without confounding effects.

## 6. Conclusions

The articles reviewed here have made remarkable advances to our knowledge regarding the role of the glial scar and each cell type comprising it. Moreover, they have proven that ablation of astrocytes, microglia, or NG2^+^-glia after SCI is not a beneficial approach to promote recovery. It is an established fact that glial cells produce inhibitory molecules at the site of the lesion, however, they also play a protective role that seems to outweigh said inhibition. Ablation of these populations all result in extended tissue damage, increased blood-borne infiltration, and reduced locomotor recovery, indicating that alternative paths should be explored for therapeutic purposes.

Among the inhibitory effects of the glial scar, the physical barrier has been pointed out as a main limitation for axon regrowth. However, Anderson et al. proved that, with the proper stimulation, axon regrowth can be achieved through and beyond the glial scar and, surprisingly, exclusively in the presence of astrocytes [21]. Moreover, for therapeutic strategies, it should be kept in mind that in human cases of SCI, the glial scar does not create such a thick wall of astrocytic processes [9], suggesting that if astrocytes inhibit neurite outgrowth in the injured human spinal cord, it is not mainly because of a physical barrier.

Alteration to the ECM composition is another grand inhibitory change after injury to the spinal cord. Glial cells have been shown to overexpress inhibitory CSPGs in the glial scar [21,22,23,24]. However, other cell types, such as pericytes and infiltrating macrophages, have been shown to also affect the ECM composition [23,118]. Therefore, it seems that reduction of the fibrotic scar might be a more beneficial approach to promote functional recovery after SCI. In fact, the inhibition of pericyte proliferation after SCI successfully reduced the fibrotic scar, resulting in improved motor and sensorimotor recovery [119].

The success of the approach by Dias et al. was based on an intermediate phenotype caused by an incomplete tamoxifen-induced recombination of the model mice [119]. They report a low number of highly recombinant mice, in which the fibrotic scar formation was completely blocked, resulting in failure to close the lesion and proving that these pericytes are essential for the generation of the fibrotic scar. In turn, the animals with an intermediate phenotype present reduced fibrotic scar density without losing tissue integrity. They also exhibit reduced inflammation, astrocyte reactivity, and density of the processes forming a wall. Curiously, reduced astrogliosis was associated with greater tissue damage in experiments eliminating microglia or NG2^+^-glia [32,34] but not in the experiment reducing pericyte proliferation [119]; possibly because the latter resulted in reduced fibrotic scarring, therefore limiting the damage extension.

Moreover, mice with a reduced fibrotic scar presented an increase in axon regrowth; CST axons growing through the glial scar do so in alignment with astrocyte processes, which might explain the astrocyte-dependent axon regrowth that was previously reported [21]. Finally, Dias et al. observed improved sensorimotor recovery after reducing the fibrotic scar [119]. These results could also help clarify the observed effects of eliminating NG2^+^-cells, which both enhanced axon regrowth and impaired locomotor recovery [34]. Considering the increase in axon regrowth after inhibition of pericyte proliferation, it is reasonable to think that this same effect after NG2^+^-cell ablation is mediated by NG2^+^-pericytes. At the same time, the opposite effects in locomotor recovery seem to indicate a protective role of NG2^+^-glia since inhibiting pericyte proliferation promotes locomotor recovery [119], but ablating NG2^+^-cells impairs it [34]. However, it would be interesting to design an approach that allows ablation of NG2^+^-glia exclusively in order to confirm its role in the glial scar without confounding effects on other cell populations.

Overall, the evidence presented here proves that modifying the glial scar by ablating cell populations is not a promising approach to promote functional recovery after injury to the spinal cord. Alternatively, a promising approach seems to be the reduction -but not elimination- of the fibrotic scar, reducing the production of ECM inhibitory components, inflammation, and astrogliosis, without preventing wound healing. It is possible that minimizing the fibrotic scar could alter the number and transcriptomic profile, not only of astrocytes but also of microglia and NG2^+^-glia. Further research on this topic could open up a new path for therapeutic strategies after spinal cord injury.

## Figures and Tables

**Figure 1 cells-12-01842-f001:**
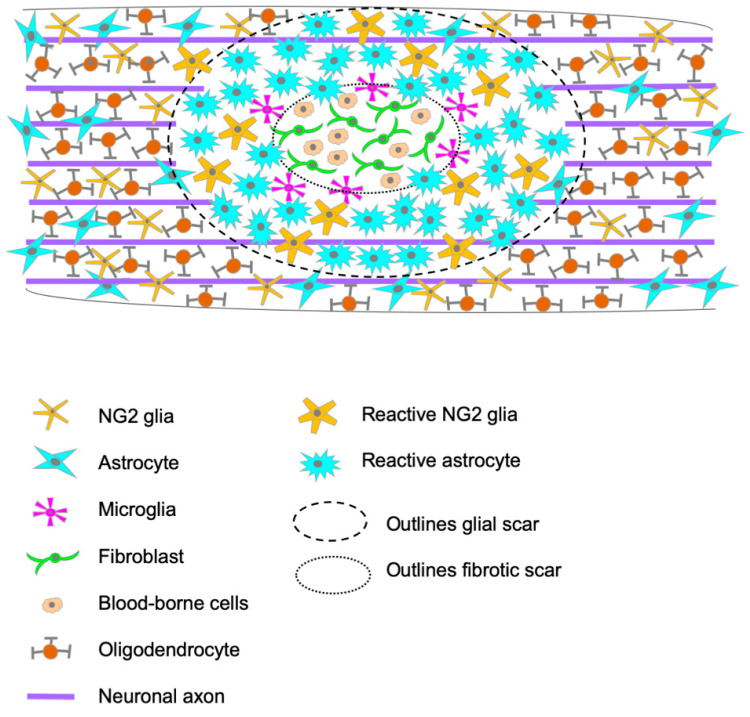
Schematic representation of the injury site, portraying the cellular composition of the glial scar and the infiltration of blood-borne cells into the spinal cord parenchyma in the epicenter of the lesion.

**Figure 2 cells-12-01842-f002:**
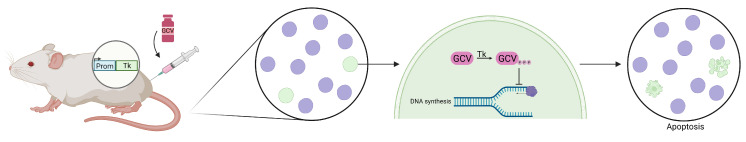
Simplified scheme of the mechanism behind the Thymidine Kinase-Ganciclovir (Tk-GCV) system for cell ablation. Purple circles represent cells that do not express the Tk transgene; green circles represent cells that express the Tk transgene under the chosen cell-specific promoter. Only the cells expressing Tk undergo apoptosis by modification of the ganciclovir prodrug to its active form. Prom—promoter; Tk—Thymidine kinase; GCV—ganciclovir.

**Table 1 cells-12-01842-t001:** Summary of the glial-cell ablation studies’ methodology and main results.

Ablated Cell Type	Cell Ablation Technique	Species/ Strain	Animal Age	Time-Point of Cell Ablation	Time-Point of Tissue Collection	Consequences of Cell Ablation	Reference
Astrocytes	*GFAP-Tk* mice. Tk-GCV system	Transgenic mice on C57BL/6J background	Not specified	from 0 to 7 dpi	14 dpi	Tissue degeneration. Increased infiltration of pro-inflammatory cells. Failed repair of the BBB. Reduced oligodendrocyte and neuronal survival. Increased demyelination. Impaired locomotor recovery.	[29]
Astrocytes	*GFAP-Tk* mice. Tk-GCV system	Transgenic mice on C57BL/6J background (females)	8–16 weeks	from 0 to 7 dpi	14–56 dpi	Lesion expansion. Unchanged axonal regrowth and extended axonal dieback. Maintained CSPGs production. Ablation of astrocytes eliminates the beneficial effect of conditioning lesions and neurotrophic factors on axonal regrowth.	[21]
	*loxP-DTR* mice. DTA-DTR system	*loxP-DTR* mice on C57BL/6 background (females)	8–16 weeks	from 35 to 45 dpi	70 dpi	Tissue degeneration and lesion expansion. Axon regrowth was not enhanced.	
Astrocytes	lentivirus-mediated Tk/GCV system	C57BL/6 mice (females)	8–12 weeks	from 1 to 8 dpi	14, 28, and 42 dpi	Enhanced infiltration of proinflammatory cells and tissue damage. Augmented neuronal death. Impaired locomotor recovery	[30]
Microglia	PLX5622 Feeding	C57BL/6N mice	8–10 weeks	from 21 d before injury to 35 dpi; from 0 to 35 dpi; from 21 d before injury to 0 dpi	1, 4, 7, 14 and 35 dpi	Disorganization of the glial scar. Enhanced infiltration of immune cells. Reduced oligodendrocyte and neuronal survival. Impaired locomotor recovery	[31]
Microglia	PLX3397 Intragastric	C57BL/6 mice (females)	8–10 weeks	14 days pre-injury until 7 or 14 dpi	7 and 14 dpi	Disorganization of the glial scar. Reduced astrocytic proliferation. Increased proinflammatory factors. Reduced neuronal survival. Impaired locomotor recovery.	[32]
Microglia	PLX3397 Feeding	C57BL/6 mice (females)	6 weeks	from 7 days before injury to 28 dpi	7, 14, 28, 30, and 56 dpi	Disorganization of the glial scar. Acute expansion of the lesion. Increased axon dieback. Impaired locomotor recovery	[33]
NG2 cells	*NG2-Tk* mice. Tk-GCV system	Transgenic mice on C57BL/6J background	12 weeks	from 0 dpi to 7 or 14 dpi	7, 14, and 21 dpi	Edema and swelling. Reduced intra-lesion laminin. Disorganization of astrocytes and the glial scar. Increased macrophage infiltration. Enhanced axon regrowth. Impaired locomotor recovery.	[34]

Tk-GCV—thymidine kinase-ganciclovir; dpi—days post-injury; DTA-DTR—diphtheria toxin subunit A-diphtheria tox.

## Data Availability

Data sharing is not applicable to this article as no new data were created or analyzed in this study.

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
