# Peer review of "Understanding the Role of the Glial Scar through the Depletion of Glial Cells after Spinal Cord Injury"

_cells, 2023, doi:10.3390/cells12141842_

Round 1

Reviewer 1 Report

In this review manuscript the authors discussed the role of glial scar (by depleting glial cells) in spinal cord injury. The topic is interesting. Some concerns and suggestions are listed as below:

Please remove extra spaces in lines 27 and 29 (lines 290 and 296). Please check throughout the manuscript.

A figure regarding glial scar formation and composition (at different time points) can be added. The spinal injury scar in fact comprises multiple cellular and extracellular components. 

The authors mentioned BBB in line 35, while this review focused on spinal cord injury.

Regarding conditional cell ablation strategies, one recent study suggested that a small portion (less than 1%) of Cx3cr1wt/wtCre−Eyfp− microglia do not carry genetic labels in the widely used Cx3cr1GFP/wt or Cx3cr1CreER-Eyfp/wt mouse models (PMID: 35062962). This point should not be ignored. Not being aware of this population may lead to significant data misinterpretation since these cells may escape detection (not carrying the Eyfp or Gfp) and cannot be modified (lacking Cre expression) as expected. 

The function of glial cells (at different time points) in spinal cord injury should be dicussed in details. Acute or chronic satge is too general. CNS microenvironment should also be considered in this case. How did you interpret inconsistent results in the literature (seemingly opposing roles)?

Some may argue that the depletion efficiency of glial cells (even using the same method) would be different in previous studies. And this may have potential effects on final results.

As a consequence of spinal cord injury, baseline microglia undergo permanent transcriptional reprogramming into a previously uncharacterized disease-associated microglia and disease-associated microglia enhance recovery of hindlimb locomotor function following injury (PMID: 34417326). Therefore, microglia and disease-associated microglia should be discussed separately since they may play different roles in the condition of spinal cord injury.

The ablation of specific glial cells using different methods may cause inflammation and then influence disease recovery.

What should be done in future studies? This paragraph should be added. Rather than focus on good versus bad, perhaps efforts would be best directed at understanding and targeting specific aspects of the scar to aid recovery. 

Emerging data has convincingly demonstrated the existence of sex-dependent structural and functional differences of rodent microglia (PMID: 33731174). Sex differences should be discussed in this case.

How about the role of glial scar in patients with SCI?

fine

Reviewer 2 Report

This is a nice easy to read and short review on the use of ablation techniques to examine the role of astrocytes and other cells in the formation of the glial scar.  The review touches on the importance of astrocyte, NG2 cells and microglia on their roles in mediating glial scar and the changes in axonal outgrowth, glial scar formation, ECM, and functional recovery after their ablation.  Although the review is somewhat comprehensive, it is missing some key papers that provide further insight into the role of pericytes and infiltrating leukocytes. 

Major Concerns:

The authors fail to cite the very first astrocyte ablation paper (Bush et al., 1999; Neuron 23:297).  This paper identifies many of the problem that occur with ablation of astrocytes and that the lack of blood brain barrier supports higher infiltration of a variety of immune cells and increased glutamate excitotoxicity, explaining many of the negative effects of these ablations.  They also demonstrate that reduced reactive astrocytes support better axonal growth.

There are no papers cited indicating the importance of pericytes in lack of regeneration and scar formation.  Although much of the paper focuses on the Anderson paper, indicated that fibroblasts (pericytes) within the lesion core are not growth inhibitory, the paper by Dias (Dias et al., 2018 Cell, 173:153-165) shows that the cells within the lesion core are inhibitory to axonal regeneration and ablation of these pericyte supports higher levels of regeneration even in the presence of astrocytes. 

The conclusions of the paper are relatively simple and do not attempt to synthesize a cohesive understanding of the dynamics between the cells and how they interact to alter scar formation, extracellular matrix and axonal growth.  They fail to explain that ablation of astrocytes leads to increased infiltration of leukocytes and increased cytokine production and that the loss of astrocytes contribute to increase excitotoxicity.  Infiltrating leukocyte also increase the amount of cytokines which regulate the production of ECMs, like NG2, tenacin, phosphacan and neurocan (Smith and Strunz 2005, Glia 52:209).  Additionally, earlier papers by Paul Reier and Jerry Silver demonstrate that astrocytes also form a physical barrier to axonal growth as demonstrated by the overlapping of astrocyte process with dense arrays of junction between them and the carpet like basal lamina that forms between astrocytes and fibroblast/pericytes at the lesion core.  Likewise, a nice paper by Bundersen et al., 2003, J. Neuroscience 23:7789) demonstrates the cellular interactions at the scar interface that organized the dense BL formation. 

Minor Concerns:

The authors only mention NG2 as a CSPG, but NG2 only as 1 or 2 CS side chains and not a very potent inhibitor of axonal growth.  Other CSPG, like versican, brevican, neurocan and phosphacan contain many more CS side chains and thus stronger inhibitors of outgrowth.

In part 3. Astrocytes in the Glial Scar….. the end of the first paragraph discussion the transformation of astrocytes from a growth permissive to a growth inhibitory state with age.  The authors should cite the original paper which describes this transformation in greater detail (Smith et al., 1986 J. Comp. Neurol. 251:23).

At the end of page 8 the last sentence in part 3:  “Hence, therapies to promote axonal regeneration can be explored in the presence of the astrocytic scar.”  Is not entirely a correct statement.  Although astrocyte ablation studies show that reactive astrocytes do play a beneficial role, they still block axonal regeneration.  The beneficial role they play in scar is to close off the injury site and protect the CNS environment from bad stuff getting in, like blood born leukocytes.  They also reduce excitotoxicity since they are the leading scavengers of glutamate after injury and in the normally functioning CNS.  Their ablation supports growth by reducing some aspects of the physical nature of the scar and reduction in BL formation – it not just dependent on CSPGs as suggested.   Likewise, most studies promoting axonal regeneration using PTEN KO or other mechanisms indicate glial bridging across the injury site are required and that as the glial scar develops the extent of axonal growth across the lesion decreases.

Reviewer 3 Report

This manuscript is a good review summarizing experimental studies of genetic and pharmacological ablation of glial cells in rodent models of spinal cord injury in order to understand the role of glial scar in recovery after spinal cord injury. The manuscript is well written and suitable for publication in Cells.

I only recommend minor editing of English language.

Round 2

Reviewer 1 Report

The authors have addressed my concerns.

fine

Reviewer 2 Report

The authors adequately responded to the reviewer's concerns.